# WHAI: WEIBULL HYBRID AUTOENCODING INFERENCE FOR DEEP TOPIC MODELING

**Hao Zhang, Bo Chen**[*] **& Dandan Guo**
National Laboraory of Radar Signal Processing,
Collaborative Innovation Center of Information Sensing and Understanding,
Xidian University, Xi'an, China.
`zhanghao_xidian@163.com`   `bchen@mail.xidian.edu.cn`
`gdd_xidian@126.com`

**Mingyuan Zhou**
McCombs School of Business,
The University of Texas at Austin, Austin, TX 78712, USA.
`Mingyuan.Zhou@mccombs.utexas.edu`

## ABSTRACT

To train an inference network jointly with a deep generative topic model, making it both scalable to big corpora and fast in out-of-sample prediction, we develop Weibull hybrid autoencoding inference (WHAI) for deep latent Dirichlet allocation, which infers posterior samples via a hybrid of stochastic-gradient MCMC and autoencoding variational Bayes. The generative network of WHAI has a hierarchy of gamma distributions, while the inference network of WHAI is a Weibull upward-downward variational autoencoder, which integrates a deterministic-upward deep neural network, and a stochastic-downward deep generative model based on a hierarchy of Weibull distributions. The Weibull distribution can be used to well approximate a gamma distribution with an analytic Kullback-Leibler divergence, and has a simple reparameterization via the uniform noise, which help efficiently compute the gradients of the evidence lower bound with respect to the parameters of the inference network. The effectiveness and efficiency of WHAI are illustrated with experiments on big corpora.

## 1 INTRODUCTION

There is a surge of research interest in multilayer representation learning for documents. To analyze the term-document count matrix of a text corpus, Srivastava et al. (2013) extend the deep Boltzmann machine (DBM) with the replicated softmax topic model of Salakhutdinov & Hinton (2009) to infer a multilayer representation with binary hidden units, but its inference network is not trained to match the true posterior (Mnih & Gregor, 2014) and the higher-layer neurons learned by DBM are difficult to visualize. The deep Poisson factor models of Gan et al. (2015) are introduced to generalize Poisson factor analysis (Zhou et al., 2012), with a deep structure restricted to model binary topic usage patterns. Deep exponential families (DEF) of Ranganath et al. (2015) construct more general probabilistic deep networks with non-binary hidden units, in which a count matrix can be factorized under the Poisson likelihood, with the gamma distributed hidden units of adjacent layers linked via the gamma scale parameters. The Poisson gamma belief network (PGBN) (Zhou et al., 2015; 2016) also factorizes a count matrix under the Poisson likelihood, but factorizes the shape parameters of the gamma distributed hidden units of each layer into the product of a connection weight matrix and the gamma hidden units of the next layer, resulting in strong nonlinearity and readily interpretable multilayer latent representations.

Those multilayer probabilistic models are often characterized by a top-down generative structure, with the distribution of a hidden layer typically acting as a prior for the layer below. Despite being able to infer a multilayer representation of a text corpus with scalable inference (Patterson &

---

[*]Corresponding author

Teh, 2013; Ruiz et al., 2016; Cong et al., 2017a), they usually rely on an iterative procedure to infer the latent representation of a new document at the testing stage, regardless of whether variational inference or Markov chain Monte Carlo (MCMC) is used. The potential need of a large number of iterations per testing document makes them unattractive when real-time processing is desired. For example, one may need to rapidly extract the topic-proportion vector of a document and use it for downstream analysis, such as identifying key topics and retrieving related documents. A potential solution is to construct a variational autoencoder (VAE) that learns the parameters of an inference network (recognition model or encoder) jointly with those of the generative model (decoder) (Kingma & Welling, 2014; Rezende et al., 2014). However, most existing VAEs rely on Gaussian latent variables, with the neural networks (NNs) acting as nonlinear transforms between adjacent layers (Sonderby et al., 2016; Dai et al., 2016; Ishaan et al., 2017). A primary reason is that there is a simple reparameterization trick for Gaussian latent variables that allows efficiently computing the noisy gradients of the evidence lower bound (ELBO) with respect to the NN parameters. Unfortunately, Gaussian based distributions often fail to well approximate the posterior distributions of sparse, nonnegative, and skewed document latent representations. For example, Srivastava & Sutton (2017) propose autoencoding variational inference for topic models (AVITM), as shown in Fig. 2b, which utilizes the logistic-normal distribution to approximate the posterior of the latent representation of a document; even though the generative model is latent Dirichlet allocation (LDA) (Blei et al., 2003), a basic single-hidden-layer topic model, due to the insufficient ability of the logistic-normal distribution to model sparsity, AVITM has to rely on some heuristic to force the latent representation of a document to be sparse. Another common shortcoming of existing VAEs is that they often only provide a point estimate for the global parameters of the generative model, and hence their inference network is optimized to approximate the posteriors of the local parameters conditioning on the data and the point estimate, rather than a full posterior, of the global parameters. In addition, from the viewpoint of probabilistic modeling, the inference network of a VAE is often merely a shallow probabilistic model, whose parameters, though, are deterministically nonlinearly transformed from the observations via a non-probabilistic deep neural network.

To address these shortcomings and move beyond Gaussian latent variable based deep models and inference procedures, we develop Weibull hybrid autoencoding inference (WHAI), a hybrid Bayesian inference for deep topic modeling that integrates both stochastic-gradient MCMC (Welling & Teh, 2011; Ma et al., 2015; Cong et al., 2017a) and a multilayer Weibull distribution based VAE. WHAI is related to a VAE in having both a decoder and encoder, but differs from a usual VAE in the following ways: 1) deep latent Dirichlet allocation (DLDA), a probabilistic deep topic model equipped with a gamma belief network, acts as the generative model; 2) inspired by the upward-downward Gibbs sampler of DLDA, as sketched in Fig. 2c, the inference network of WHAI uses a upward-downward structure, as shown in Fig. 2a, to combine a non-probabilistic bottom-up deep NN and a probabilistic top-down deep generative model, with the $\ell$th hidden layer of the generative model linked to both the $(\ell+1)$th hidden layer of itself and the $\ell$th hidden layer of the deep NN; 3) a hybrid of stochastic-gradient MCMC and autoencoding variational inference is employed to infer both the posterior distribution of the global parameters, represented as collected posterior MCMC samples, and a VAE that approximates the posterior distribution of the local parameters given the data and a posterior sample (rather than a point estimate) of the global parameters; 4) we use the Weibull distributions in the inference network to approximate gamma distributed conditional posteriors, exploiting the fact that the Weibull and gamma distributions have similar probability density functions (PDFs), the Kullback-Leibler (KL) divergence from the Weibull to gamma distributions is analytic, and a Weibull random variable can be efficiently reparameterized with a uniform noise.

Note that we have also tried gamma hybrid autoencoding inference (GHAI), which directly uses the gamma distribution in the probabilistic top-down part of the inference network, while using rejection sampling variational inference (RSVI) of Naesseth et al. to approximately compute the gradient of the ELBO. While RSVI is a very general technique that can be applied to a wide variety of non-reparameterizable distributions, we find that for replacing the reparameterizable Weibull with non-reparameterizable gamma distributions in the inference network, the potential gains are overshadowed by the disadvantages of having to rely on an approximate reparameterization scheme guided by rejection sampling. In the experiments for deep topic modeling, we show that WHAI clearly outperforms GHAI, and both WHAI and GHAI outperform their counterparts that remove the top-down links of the inference network, referred to as WHAI-independent and GHAI-independent, respectively; WHAI is comparable to Gibbs sampling in terms performance, but is scalable to big

training data via mini-batch stochastic-gradient based inference and is considerably fast in out-of-sample prediction via the use of an inference network.

## 2 WHAI FOR MULTILAYER DOCUMENT REPRESENTATION

Below we first describe the decoder and encoder of WHAI, and then provide a hybrid stochastic-gradient MCMC and autoencoding variational inference that is fast in both training and testing.

### 2.1 DOCUMENT DECODER: DEEP LATENT DIRICHLET ALLOCATION

In order to capture the hierarchical document latent representation, WHAI uses the Poisson gamma belief network (PGBN) of Zhou et al. (2016), a deep probabilistic topic model, as the generative network (encoder). Choosing a deep generative model as its decoder distinguishes WHAI from both AVITM, which uses a "shallow" LDA as its decoder, and a conventional VAE, which often uses as its decoder a "shallow" (transformed) Gaussian distribution, whose parameters are deterministically nonlinearly transformed from the observation via "black-box" deep neural networks. With all the gamma latent variables marginalized out, as shown in Cong et al. (2017a), the PGBN can also be represented as deep LDA (DLDA). For simplicity, below we use DLDA to refer to both the PGBN and DLDA representations of the same underlying deep generative model, as briefly described below. Note the single-hidden-layer version of DLDA reduces to Poisson factor analysis of Zhou et al. (2012), which is closely related to LDA. Let us denote $\mathbf{\Phi}^{(1)} \in \mathbb{R}_+^{K_0 \times K_1}$ and $\boldsymbol{\theta}_n^{(1)} \in \mathbb{R}_+^{K_1}$ as the factor loading and latent representation of the first hidden layer of DLDA, respectively, where $\mathbb{R}_+ = \{x, x \geq 0\}$ and $K_1$ is the number of topics (factors) of the first layer. We further restrict that the sum of each column of $\mathbf{\Phi}^{(1)}$ is equal to one. To model high-dimensional multivariate sparse count vectors $\boldsymbol{x}_n \in \mathbb{Z}^{K_0}$, where $\mathbb{Z} = \{0, 1, \ldots\}$, under the Poisson likelihood, the DLDA generative model with $L$ hidden layers, from top to bottom, can be expressed as

$$\boldsymbol{\theta}_n^{(L)} \sim \text{Gam}\left(\boldsymbol{r}, c_n^{(L+1)}\right), \ldots, \boldsymbol{\theta}_n^{(l)} \sim \text{Gam}\left(\mathbf{\Phi}^{(l+1)}\boldsymbol{\theta}_n^{(l+1)}, c_n^{(l+1)}\right), \ldots,$$

$$\boldsymbol{\theta}_n^{(1)} \sim \text{Gam}\left(\mathbf{\Phi}^{(2)}\boldsymbol{\theta}_n^{(2)}, c_n^{(2)}\right), \; \boldsymbol{x}_n \sim \text{Pois}\left(\mathbf{\Phi}^{(1)}\boldsymbol{\theta}_n^{(1)}\right). \tag{1}$$

where the hidden units $\boldsymbol{\theta}_n^{(l)} \in \mathbb{R}_+^{K_l}$ of layer $l$ are factorized into the product of the factor loading $\mathbf{\Phi}^{(l)} \in \mathbb{R}_+^{K_{l-1} \times K_l}$ and hidden units of the next layer. It infers a multilayer data representation, and can visualize its topic $\phi_k^{(l)}$ at hidden layer $l$ as $\left[\prod_{t=1}^{l-1} \mathbf{\Phi}^{(t)}\right] \phi_k^{(l)}$, which tend to be very specific in the bottom layer and become increasingly more general when moving upward. The unsupervisedly extracted multilayer latent representations $\boldsymbol{\theta}_n^{(l)}$ are well suited for additional downstream analysis, such as document classification and retrieval.

The upward-downward Gibbs sampling for DLDA, as described in detail in Zhou et al. (2016), is sketched in Fig. 2c, where $\mathbf{Z}^l$ represent augmented latent counts that are sampled upward given the observations and model parameters. While having closed-form update equations, the Gibbs sampler requires processing all documents in each iteration and hence has limited scalability. Consequently, a topic-layer-adaptive stochastic gradient Riemannian (TLASGR) MCMC for DLDA, referred to as DLDA-TLASGR, is proposed to process big corpora (Cong et al., 2017a). Different from AVITM (Srivastava & Sutton, 2017) that models a probabilistic simplex with the expanded-natural representation (Patterson & Teh, 2013), DLDA-TLASGR uses a more elegant simplex constraint and increases the sampling efficiency via the use of the Fisher information matrix (FIM) (Cong et al., 2017a;b), with adaptive step-sizes for the topics of different layers. Specifically, suppose $\phi_k^{(l)}$ is the $k$th topic in layer $\ell$ with prior $\phi_k^{(l)} \sim \text{Dirichlet}(\eta_k^{(l)})$, sampling it can be efficiently realized as

$$(\phi_k)_{t+1} = \left[(\phi_k)_t + \frac{\varepsilon_t}{M_k}\left[(\rho \tilde{\boldsymbol{z}}_{:k\cdot} + \eta_k^{(l)}) - (\rho \tilde{z}_{\cdot k\cdot} + \eta_k^{(l)} V)(\phi_k)_t\right] + \mathcal{N}\left(\mathbf{0}, \frac{2\varepsilon_t}{M_k}diag(\phi_k)_t\right)\right]_{\angle}, \quad (2)$$

where $M_k$ is calculated using the estimated FIM, both $\tilde{\boldsymbol{z}}_{:k\cdot}$ and $\tilde{z}_{\cdot k\cdot}$ come from the augmented latent counts $\mathbf{Z}$, and $[\cdot]_{\angle}$ denotes a simplex constraint; more details about TLASGR-MCMC for DLDA can be found in Cong et al. (2017a) and are omitted here for brevity.

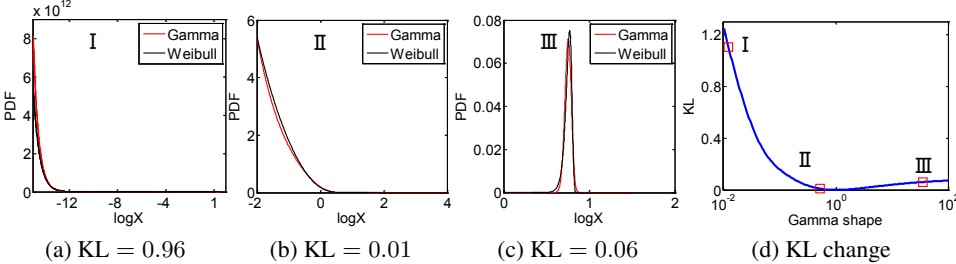

Figure 1: The KL divergence from the inferred Weibull distribution to the target gamma one as (a) Gamma$(0.05, 1)$, (b) Gamma$(0.5, 1)$, and (c) Gamma$(5, 1)$. Subplot (d) shows the KL divergence as a function of the gamma shape parameter, where the gamma scale parameter is fixed at 1.

Despite the attractive properties, neither the Gibbs sampler nor TLASGR-MCMC of DLDA can avoid taking a potentially large number of MCMC iterations to infer the latent representation of a testing document, which hinders real-time processing of the incoming documents and motivates us to construct an inference network with fast out-of-sample prediction, as described below.

## 2.2 DOCUMENT ENCODER: WEIBULL UPWARD-DOWNWARD VARIATIONAL ENCODER

A VAE uses an inference network to map the observations directly to their latent representations. However, their success so far is mostly restricted to Gaussian distributed latent variables, and does not generalize well to model sparse, nonnegative, and skewed latent document representations. To move beyond latent Gaussian models, below we propose Weibull upward-downward variational encoder (WUDVE) to efficiently produce a document's multilayer latent representation under DLDA.

Assuming the global parameters $\phi_k^{(l)}$ of DLDA shown in (1) are given and the task is to infer the local parameters $\boldsymbol{\theta}_n^{(l+1)}$, the usual strategy of mean-field variational Bayes (Jordan et al., 1999) is to maximize the ELBO that can be expressed as

$$L = \sum_{n=1}^{N} \mathbb{E}\left[\ln p\left(\boldsymbol{x}_n \,|\, \boldsymbol{\Phi}^{(1)}, \boldsymbol{\theta}_n^{(1)}\right)\right] - \sum_{n=1}^{N}\sum_{l=1}^{L} \mathbb{E}\left[\ln \frac{q\left(\boldsymbol{\theta}_n^{(l)}\right)}{p\left(\boldsymbol{\theta}_n^{(l)} \,|\, \boldsymbol{\Phi}^{(l+1)}, \boldsymbol{\theta}_n^{(l+1)}\right)}\right], \tag{3}$$

where the expectations are taken with respect to (w.r.t.) a fully factorized distribution as

$$q\left(\{\boldsymbol{\theta}_n^{(l)}\}_{n=1,l=1}^{N,L}\right) = \prod_{n=1}^{N}\prod_{l=1}^{L} q\left(\boldsymbol{\theta}_n^{(l)}\right). \tag{4}$$

Instead of using a conventional latent Gaussian based VAE, in order to model sparse and nonnegative latent document representation, it might be more appropriate to use a gamma distribution based inference network defined as $q(\boldsymbol{\theta}_n \,|\, \boldsymbol{x}_n) = \text{Gamma}(f_{\mathbf{W}}(\boldsymbol{x}_n), g_{\mathbf{W}}(\boldsymbol{x}_n))$, where $f$ and $g$ are two related deep neural networks parameterized by $\mathbf{W}$. However, it is hard to efficiently compute the gradient of the ELBO with respect to $\mathbf{W}$, due to the difficulty to reparameterize a gamma distributed random variable (Kingma & Welling, 2014; Ruiz et al., 2016; Knowles, 2015), motivating us to identify a surrogate distribution that can not only well approximate the gamma distribution, but also be easily reparameterized. Below we show the Weibull distribution is an ideal choice.

### 2.2.1 WEIBULL AND GAMMA DISTRIBUTIONS

A main reason that we choose the Weibull distribution to construct the inference network is that the Weibull and gamma distributions have similar PDFs:

$$\text{Weibull PDF: } P(x \,|\, k, \lambda) = \frac{k}{\lambda^k} x^{k-1} e^{(x/\lambda)^k}, \quad \text{Gamma PDF: } P(x \,|\, \alpha, \beta) = \frac{\beta^\alpha}{\Gamma(\alpha)} x^{\alpha-1} e^{-\beta x},$$

where $x \in \mathbb{R}_+$. Another reason is due to a simple reparameterization for $x \sim \text{Weibull}(k, \lambda)$ as

$$x = \lambda(-\ln(1-\epsilon))^{1/k}, \ \epsilon \sim \text{Uniform}(0, 1).$$

Moreover, its KL-divergence from the gamma distribution has an analytic expression as

$$KL(\text{Weibull}(k,\lambda)||\text{Gamma}(\alpha,\beta)) = \alpha \ln \lambda - \frac{\gamma \alpha}{k} - \ln k - \beta \lambda \Gamma\left(1 + \frac{1}{k}\right) + \gamma + 1 + \alpha \ln \beta - \ln \Gamma(\alpha).$$

Minimizing this KL divergence, one can identify the two parameters of a Weibull distribution to approximate a given gamma one. As shown in Fig. 1, the inferred Weibull distribution in general quite accurately approximates the target gamma one, as long as the gamma shape parameter is neither too close to zero nor too large.

### 2.2.2 UPWARD-DOWNWARD INFORMATION PROPAGATION

For the DLDA upward-downward Gibbs sampler sketched in Fig. 2c, the corresponding Gibbs sampling update equation for $\boldsymbol{\theta}_n^{(l)}$ can be expressed as

$$(\boldsymbol{\theta}_n^{(l)} \,|\, -) \sim \text{Gamma}\left(\boldsymbol{m}_n^{(l)(l+1)} + \boldsymbol{\Phi}^{(l+1)}\boldsymbol{\theta}_n^{(l+1)}, f(p_n^{(l)}, c_n^{(l+1)})\right), \tag{5}$$

where $\boldsymbol{m}_n^{(l)(l+1)}$ and $p_n^{(l)}$ are latent random variables constituted by information upward propagated to layer $l$, as described in detail in Zhou et al. (2016) and hence omitted here for brevity. It is clear from (5) that the conditional posterior of $\boldsymbol{\theta}_n^{(l)}$ is related to both the information at the higher (prior) layer, and that upward propagated to the current layer via a series of data augmentation and marginalization steps described in Zhou et al. (2016). Inspired by this instructive upward-downward information propagation in Gibbs sampling, as shown in Fig. 2a, we construct WUDVE, the inference network of our model, as $q(\boldsymbol{\theta}_n^{(L)} \,|\, \boldsymbol{h}_n^{(L)}) \prod_{l=1}^{L-1} q(\boldsymbol{\theta}_n^{(l)} \,|\, \boldsymbol{\Phi}^{(l+1)}, \boldsymbol{h}_n^{(l)}, \boldsymbol{\theta}_n^{(l+1)})$, where

$$q(\boldsymbol{\theta}_n^{(l)} \,|\, \boldsymbol{\Phi}^{(l+1)}, \boldsymbol{h}_n^{(l)}, \boldsymbol{\theta}_n^{(l+1)}) = \text{Weibull}(\boldsymbol{k}_n^{(l)} + \boldsymbol{\Phi}^{(l+1)}\boldsymbol{\theta}_n^{(l+1)}, \boldsymbol{\lambda}_n^{(l)}). \tag{6}$$

The Weibull distribution is used to approximate the gamma distributed conditional posterior, and its parameters $\boldsymbol{k}_n^{(l)} \in \mathbb{R}^{K_l}$ and $\boldsymbol{\lambda}_n^{(l)} \in \mathbb{R}^{K_l}$ are both deterministically transformed from the observation $\boldsymbol{x}_n$ using the neural networks, as illustrated in Fig. 2a and specified as

$$\boldsymbol{k}_n^{(l)} = \ln[1 + \exp(\mathbf{W}_1^{(l)}\boldsymbol{h}_n^{(l)} + \boldsymbol{b}_1^{(l)})], \tag{7}$$

$$\boldsymbol{\lambda}_n^{(l)} = \ln[1 + \exp(\mathbf{W}_2^{(l)}\boldsymbol{h}_n^{(l)} + \boldsymbol{b}_2^{(l)})], \tag{8}$$

$$\boldsymbol{h}_n^{(l)} = \ln[1 + \exp(\mathbf{W}_3^{(l)}\boldsymbol{h}_n^{(l-1)} + \boldsymbol{b}_3^{(l)})], \tag{9}$$

where $\boldsymbol{h}_n^{(0)} = \log(1 + \boldsymbol{x}_n)$, $\mathbf{W}_1^{(l)} \in \mathbb{R}^{K_l \times K_l}$, $\mathbf{W}_2^{(l)} \in \mathbb{R}^{K_l \times K_l}$, $\mathbf{W}_3^{(l)} \in \mathbb{R}^{K_l \times K_{l-1}}$, $\boldsymbol{b}_1^{(l)} \in \mathbb{R}^{K_l}$, $\boldsymbol{b}_2^{(l)} \in \mathbb{R}^{K_l}$, and $\boldsymbol{b}_3^{(l)} \in \mathbb{R}^{K_l}$. This upward-downward inference network is distinct from that of a usual VAE, where it is common that the inference network has a pure bottom-up structure and only interacts with the generative model via the ELBO (Kingma & Welling, 2014; Ishaan et al., 2017). Note that WUDVE no longer follows mean-field variational Bayes to make a fully factorized assumption as in (4).

Comparing Figs. 2c and 2a show that in each iteration, both Gibbs sampling and WUDVE have not only an upward information propagation (orange arrows), but also a downward one (blue arrows), but their underlying implementations are distinct from each other. Gibbs sampling in Fig. 2c does not have an inference network and needs the local variables $\boldsymbol{\theta}_n^{(l)}$ to help perform stochastic upward information propagation, whereas WUDVE in Fig. 2a uses its non-probabilistic part to perform deterministic upward information propagation, without relying on the local variables $\boldsymbol{\theta}_n^{(l)}$. It is also interesting to notice that the upward-downward structure of WUDVE, motivated by the upward-downward Gibbs sampler of DLDA, is closely related to that used in the ladder VAE of Sonderby et al. (2016). However, to combine the bottom-up and top-down information, ladder VAE relies on some heuristic restricted to Gaussian latent variables.

### 2.3 HYBRID MCMC/VAE INFERENCE

In Section 2.1, we describe how to use TLASGR-MCMC of Cong et al. (2017a), a stochastic-gradient MCMC algorithm for DLDA, to sample the global parameters $\{\boldsymbol{\Phi}^{(l)}\}_{1,L}$; whereas in Section 2.2.2, we describe how to use WUDVE, an autoencoding variational inference network, to

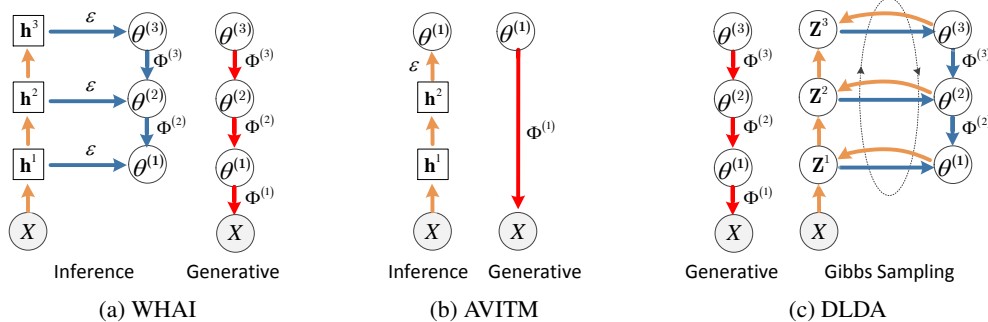

Figure 2: (a-b): Inference (or encoder/recognition) and generative (or decoder) models for (a) WHAI and (b) AVITM; (c) the generative model and a sketch of the upward-downward Gibbs sampler of DLDA, where $\mathbf{Z}^l$ are augmented latent counts that are upward sampled in each Gibbs sampling iteration. Circles are stochastic variables and squares are deterministic variables. The orange and blue arrows denote the upward and downward information propagation respectively, and the red ones denote the data generation.

approximate the conditional posterior of the local parameters $\{\boldsymbol{\theta}_n^{(l)}\}_{1,L}$ given $\{\boldsymbol{\Phi}^{(l)}\}_{1,L}$ and observation $\boldsymbol{x}_n$. Rather than merely finding a point estimate of the global parameters $\{\boldsymbol{\Phi}^{(l)}\}_{1,L}$, we describe in Algorithm 1 how to combine TLASGR-MCMC and the proposed WUDVE into a hybrid MCMC/VAE inference algorithm, which infers posterior samples for both the global parameters $\{\boldsymbol{\Phi}^{(l)}\}_{1,L}$ of the generative network, and the corresponding neural network parameters $\boldsymbol{\Omega} = \{\mathbf{W}_1^{(l)}, \boldsymbol{b}_1^{(l)}, \mathbf{W}_2^{(l)}, \boldsymbol{b}_2^{(l)}, \mathbf{W}_3^{(l)}, \boldsymbol{b}_3^{(l)}\}_{1,L}$ of the inference network. Being able to efficiently evaluating the gradient of the ELBO is important to the success of a variational inference algorithm (Hoffman et al., 2013; Paisley et al., 2012; Kingma & Welling, 2014; Mnih & Gregor, 2014; Ranganath et al., 2015; Ruiz et al., 2016; Rezende et al., 2014). An important step of Algorithm 1 is calculating the gradient of the ELBO in (3) with respect to the NN parameters $\boldsymbol{\Omega}$. Thanks to the choice of the Weibull distribution, the second term of the ELBO in (3) is analytic, and due to simple reparameterization of the Weibull distribution, the gradient of the first term of the ELBO with respect to $\boldsymbol{\Omega}$ can be accurately evaluated, achieving satisfactory performance using even a single Monte Carlo sample, as shown in our experimental results. Thanks to the architecture of WUDVE, using the inference network, for a new mini-batch, we can directly find the conditional posteriors of $\{\boldsymbol{\theta}_n^{(l)}\}_{1,L}$ given $\{\boldsymbol{\Phi}^{(l)}\}_{1,L}$ and the stochastically updated $\boldsymbol{\Omega}$, with which we can sample the local parameters and then use TLASGR-MCMC to stochastically update the global parameters $\{\boldsymbol{\Phi}^{(l)}\}_{1,L}$.

## 2.4 Variations of WHAI

To clearly understand how each component contributes to the overall performance of WHAI, below we consider two different variations: GHAI and WAI. We first consider gamma hybrid autoencoding inference (GHAI). In WUDVE, the inference network for WHAI, we have a deterministic-upward and stochastic-downward structure, where the reparameterizable Weilbull distribution is used to connect adjacent stochastic layers. Although we choose to use the Weibull distribution for the reasons specified in Section 2.2.1, one may also choose some other distribution in the downward structure. For example, one may choose the gamma distribution and replace (6) with

$$q(\boldsymbol{\theta}_n^{(l)} \mid \boldsymbol{\Phi}^{(l+1)}, \boldsymbol{h}_n^{(l)}, \boldsymbol{\theta}_n^{(l+1)}) = \text{Gamma}(\boldsymbol{k}_n^{(l)} + \boldsymbol{\Phi}^{(l+1)}\boldsymbol{\theta}_n^{(l+1)}, \boldsymbol{\lambda}_n^{(l)}). \qquad (10)$$

Even though the gamma distribution does not have a simple reparameteriation, one may use the RSVI of Naesseth et al. to define an approximate reparameterization procedure via rejection sampling. More specifically, following Naesseth et al., to generate a gamma random variable $z \sim \text{Gamma}(\alpha, \beta)$, one may first use the rejection sampler of Marsaglia & Tsang (2000) to generate $\tilde{z} \sim \text{Gamma}(\alpha + B, 1)$, for which the proposal distribution is expressed as

$$\tilde{z} = \left(\alpha + B - \frac{1}{3}\right)\left(1 + \frac{\varepsilon}{\sqrt{9(\alpha + B) - 3}}\right)^3, \quad \varepsilon \sim \mathcal{N}(0, 1),$$

---

**Algorithm 1** Hybrid stochastic-gradient MCMC and autoencoding variational inference for WHAI

Set mini-batch size $m$ and the number of layer $L$

Initialize encoder parameter $\mathbf{\Omega}$ and model parameter $\{\mathbf{\Phi}^{(l)}\}_{1,L}$.

**for** $iter = 1, 2, \cdots$ **do**

    Randomly select a mini-batch of $m$ documents to form a subset $\mathbf{X} = \{\boldsymbol{x}_i\}_{1,m}$;

    Draw random noise $\{\varepsilon_i^l\}_{i=1,l=1}^{m,L}$ from uniform distribution;

    Calculate $\nabla_{\mathbf{\Omega}} L\left(\mathbf{\Omega}, \mathbf{\Phi}^{\{l\}}; \mathbf{X}, \varepsilon_i^l\right)$ according to (3), and update $\mathbf{\Omega}$;

    Sample $\boldsymbol{\theta}_i^{\{l\}}$ from (6) via $\mathbf{\Omega}$ to update topics $\{\mathbf{\Phi}^{(l)}\}_{l=1}^{L}$ according to (2);

**end for**

---

where $B$ is a pre-set integer to make the acceptance probability be close to 1; one then lets $z = \beta^{-1}\tilde{z}\prod_{i=1}^{B} u_i^{1/(\alpha+i-1)}$, where $u_i \sim \text{Uniform}(0,1)$. The gradients of the ELBO, however, could still suffer from relatively high variance, as how likely a proposed $\varepsilon$ will be accepted depends on the gamma distribution parameters, and $B$ extra uniform random variables $\{u_i\}_{1,B}$ need to be introduced.

To demonstrate the advantages of the proposed hybrid inference for WHAI, which infers posterior samples of the global parameters, including $\{\mathbf{\Phi}^{(l)}\}_{1,L}$ and $\mathbf{\Omega}$, using TLASGR-MCMC, we also consider Weibull autoencoding inference (WAI) that has the same inference network as WHAI but infers $\{\mathbf{\Phi}^{(l)}\}_{1,L}$ and $\mathbf{\Omega}$ using stochastic gradient decent (SGD) (Kingma & Ba, 2015). Note that as argued in Mandt et al. (2017), SGD can also be used for approximate Bayesian inference. We will show in experiments that sampling the global parameters via TLASGR-MCMC provides improved performance in comparison to sampling them via SGD.

To understand the importance of the stochastic-downward structure used in the inference network, and further understand the differences between using the Weibull distribution with simple reparameterization and using the gamma distribution with RSVI, we also consider DLDA-GHAI-Independent and DLDA-WHAI-Independent that remove the stochastic-downward connections of DLDA-GHAI and DLDA-WHAI, respectively. More specifically, they define $q(\boldsymbol{\theta}_n^{(l)} \mid \mathbf{\Phi}^{(l+1)}, \boldsymbol{h}_n^{(l)}, \boldsymbol{\theta}_n^{(l+1)})$ in (6) as Weilbull$(\boldsymbol{k}_n^{(l)}, \boldsymbol{\lambda}_n^{(l)})$ and Gamma$(\boldsymbol{k}_n^{(l)}, \boldsymbol{\lambda}_n^{(l)})$, respectively, and use variational inference and RSVI, respectively, to infer $\mathbf{\Omega}$.

## 3 EXPERIMENTAL RESULTS

We compare the performance of different algorithms on 20Newsgroups (20News), Reuters Corpus Volume I (RCV1), and Wikipedia (Wiki). 20News consists of 18,845 documents with a vocabulary size of 2,000. RCV1 consists of 804,414 documents with a vocabulary size of 10,000. Wiki, with a vocabulary size of 7,702, consists of 10 million documents randomly downloaded from Wikipedia using the script provided for Hoffman et al. (2010). Similar to Cong et al. (2017a), we randomly select 100,000 documents for testing. To be consistent with previous settings (Gan et al., 2015; Henao et al., 2015; Cong et al., 2017a), no precautions are taken in the Wikipedia downloading script to prevent a testing document from being downloaded into a mini-batch for training. Our code is written in Theano (Theano Development Team, 2016).

For comparison, we consider the deep Poisson factor analysis (DPFA) of Gan et al. (2015), DLDA-Gibbs of Zhou et al. (2016), DLDA-TLASGR of Cong et al. (2017a), and AVITM of Srivastava & Sutton (2017), using the code provided by the authors. Note that as shown in Cong et al. (2017a), DLDA-Gibbs and DLDA-TLASGR are state-of-the-art topic modeling algorithms that clearly outperform a large number of previously proposed ones, such as the replicated softmax of Salakhutdinov & Hinton (2009) and the nested Hierarchical Dirichlet process of Paisley et al. (2015).

### 3.1 PER-HELDOUT-WORD PERPLEXITY

Per-heldout-word perplexity is a widely-used performance measure. Similar to Wallach et al. (2009), Paisley et al. (2011), and Zhou et al. (2012), for each corpus, we randomly select 70% of the word

Table 1: Comparison of per-heldout-word perplexity and testing time (average seconds per document) on three different datasets.

| Model | Size | Perplexity | | | Test Time | | |
|---|---|---|---|---|---|---|---|
| | | 20News | RCV1 | Wiki | 20News | RCV1 | Wiki |
| DLDA-Gibbs | 128-64-32 | 571 | 938 | 966 | 10.46 | 23.38 | 23.69 |
| DLDA-Gibbs | 128-64 | 573 | 942 | 968 | 8.73 | 18.50 | 19.79 |
| DLDA-Gibbs | 128 | 584 | 951 | 981 | 4.69 | 12.57 | 13.31 |
| DLDA-TLASGR | 128-64-32 | 579 | 950 | 978 | 10.46 | 23.38 | 23.69 |
| DLDA-TLASGR | 128-64 | 581 | 955 | 979 | 8.73 | 18.50 | 19.79 |
| DLDA-TLASGR | 128 | 590 | 963 | 993 | 4.69 | 12.57 | 13.31 |
| DPFA | 128-64-32 | 637 | 1041 | 1056 | 20.12 | 34.21 | 35.41 |
| AVITM | 128 | 654 | 1062 | 1088 | 0.23 | 0.68 | 0.80 |
| *DLDA-GHAI-Independent* | 128-64-32 | 613 | 970 | 999 | 0.62 | 1.22 | 1.47 |
| *DLDA-GHAI-Independent* | 128-64 | 614 | 970 | 1000 | 0.41 | 0.94 | 1.01 |
| *DLDA-GHAI-Independent* | 128 | 615 | 972 | 1003 | 0.22 | 0.69 | 0.80 |
| *DLDA-GHAI* | 128-64-32 | 604 | 963 | 994 | 0.66 | 1.25 | 1.49 |
| *DLDA-GHAI* | 128-64 | 608 | 965 | 997 | 0.44 | 0.96 | 1.05 |
| *DLDA-GHAI* | 128 | 615 | 972 | 1003 | 0.22 | 0.69 | 0.80 |
| *DLDA-WHAI-Independent* | 128-64-32 | 588 | 964 | 990 | 0.58 | 1.15 | 1.38 |
| *DLDA-WHAI-Independent* | 128-64 | 589 | 965 | 992 | 0.38 | 0.87 | 0.97 |
| *DLDA-WHAI-Independent* | 128 | 592 | 966 | 996 | 0.20 | 0.66 | 0.78 |
| *DLDA-WAI* | 128-64-32 | 581 | 954 | 984 | 0.63 | 1.20 | 1.43 |
| *DLDA-WAI* | 128-64 | 583 | 958 | 986 | 0.42 | 0.91 | 1.02 |
| *DLDA-WAI* | 128 | 593 | 967 | 999 | 0.20 | 0.66 | 0.78 |
| *DLDA-WHAI* | 128-64-32 | 581 | 953 | 980 | 0.63 | 1.20 | 1.43 |
| *DLDA-WHAI* | 128-64 | 582 | 957 | 982 | 0.42 | 0.91 | 1.02 |
| *DLDA-WHAI* | 128 | 591 | 965 | 996 | 0.20 | 0.66 | 0.78 |

tokens from each document to form a training matrix $\mathbf{T}$, holding out the remaining 30% to form a testing matrix $\mathbf{Y}$. We use $\mathbf{T}$ to train the model and calculate the per-heldout-word perplexity as

$$\exp\left\{-\frac{1}{y_{..}}\sum_{v=1}^{V}\sum_{n=1}^{N}y_{vn}\ln\frac{\sum_{s=1}^{S}\sum_{k=1}^{K^1}\phi_{vk}^{(1)s}\theta_{kn}^{(1)s}}{\sum_{s=1}^{S}\sum_{v=1}^{V}\sum_{k=1}^{K^1}\phi_{vk}^{(1)s}\theta_{kn}^{(1)s}}\right\}, \tag{11}$$

where $S$ is the total number of collected samples and $y_{..} = \sum_{v=1}^{V}\sum_{n=1}^{N'}y_{vn}$. For the proposed model, we set the mini-batch size as 200, and use as burn-in 2000 mini-batches for both 20News and RCV1 and 3500 for wiki. We collect 3000 samples after burn-in to calculate perplexity. The hyperparameters of WHAI are set as: $\eta^{(l)} = 1/K_l$, $\boldsymbol{r} = \mathbf{1}$, and $c_n^{(l)} = 1$.

Table 1 lists for various algorithms both the perplexity and the average run time per testing document given a single sample (estimate) of the global parameters. Clearly, given the same generative network structure, DLDA-Gibbs performs the best in terms of predicting heldout word tokens, which is not surprising as this batch algorithm can sample from the true posteriors given enough Gibbs sampling iterations. DLDA-TLASGR is a mini-batch algorithm that is much more scalable in training than DLDA-Gibbs, at the expense of slighted degraded performance in out-of-sample prediction. Both DLDA-WAI, using SGD to infer the global parameters, and DLDA-WHAI, using a stochastic-gradient MCMC to infer the global parameters, slightly underperform DLDA-TLASGR; all mini-batch based algorithms are scalable to a big training corpus, but due to the use of the WUDVE inference network, both DLDA-GHAI and DLDA-WHAI, as well as their variations, are considerably fast in processing a testing document. In terms of perplexity, all algorithms with DLDA as the generative model clearly outperform both DPFA of Gan et al. (2015) and AVITM of Srivastava & Sutton (2017), while in terms of the computational cost for testing, all algorithms with an inference network, such as AVITM, DLDA-GHAI, and DLDA-WHAI, clearly outperform these relying on an interactive procedure for out-of-sample prediction, including DPFA, DLDA-Gibbs, and DLDA-TLASGR. It is also clear that except for DLDA-GHAI-Independent and DLDA-WHAI-Independent that have no stochastic-downward components in their inference, all the other algorithms with DLDA as the generative model have a clear trend of improvement as the generative network becomes deeper, in-

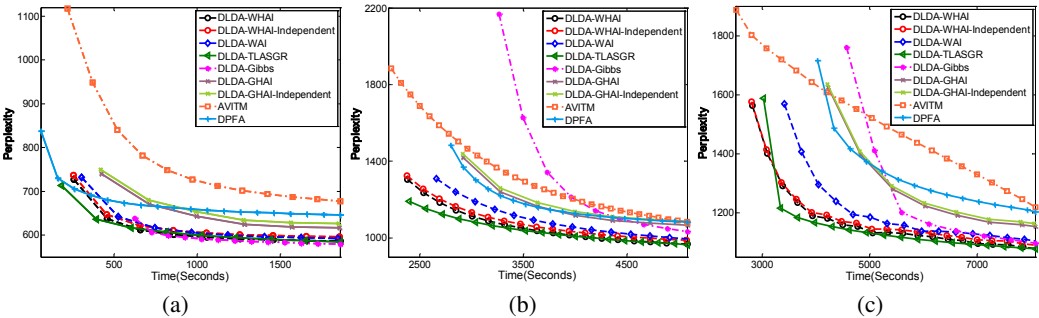

Figure 3: Plot of per-heldout-word perplexity as a function of time for (a) 20News, (b) RCV1, and (c) Wiki. Except for AVITM that has a single hidden layer with 128 topics, all the other algorithms have the same network size of 128-64-32 for their deep generative models.

dicating the importance of having stochastic-downward information propagation during posterior inference; and DLDA-WHAI with a single hidden layer already clearly outperforms AVITM, indicating that using the Weibull distribution is more appropriate than using the logistic-normal distribution to model the document latent representation. Furthermore, thanks to the use of the stochastic gradient based TLASGR-MCMC rather than a simple SGD procedure, DLDA-WHAI consistently outperforms DLDA-WAI. Last but not least, while DLDA-GHAI that relies on RSVI to approximately reparameterize the gamma distributions clearly outperforms AVITM and DPFA, it clearly underperforms DLDA-WHAI that has simple reparameterizations for its Weibull distributions.

Below we examine how various inference algorithms progress over time during training, evaluated with per-holdout-word perplexity. As clearly shown in Fig. 3, DLDA-WHAI outperforms DPFA and AVITM in providing lower perplexity as time progresses, which is not surprising as the DLDA multilayer generative model is good at document representation, while AVITM is only "deep" in the deterministic part of its inference network and DPFA is restricted to model binary topic usage patterns via its deep network. When DLDA is used as the generative model, in comparison to Gibbs sampling and TLASGR-MCMC on two large corpora, RCV1 and Wiki, the mini-batch based WHAI converges slightly slower than TLASGR-MCMC but much faster than Gibbs sampling; WHAI consistently outperforms WAI, which demonstrates the advantage of the hybrid MCMC/VAE inference; in addition, the RSVI based DLDA-GHAI clearly converges more slowly in time than DLDA-WHAI. Note that for all three datasets, the perplexity of TLASGR decreases at a fast rate, followed by closely by WHAI, while that of Gibbs sampling decreases slowly, especially for RCV1 and Wiki, as shown in Figs. 3(b-c). This is expected as both RCV1 and Wiki are much larger corpora, for which a mini-batch based inference algorithm can already make significant progress in inferring the global model parameters, before a batch-learning Gibbs sampler finishes a single iteration that needs to go through all documents. We also notice that although AVITM is fast for testing via the use of a VAE, its representation power is limited due to not only the use of a shallow topic model, but also the use of a latent Gaussian based inference network that is not naturally suited to model document latent representation.

## 3.2 TOPIC HIERARCHY AND MANIFOLD

In addition to quantitative evaluations, we have also visually inspected the inferred topics at different layers and the inferred connection weights between the topics of adjacent layers. Distinct from many existing deep learning models that build nonlinearity via "black-box" neural networks, we can easily visualize the whole stochastic network, whose hidden units of layer $l-1$ and those of layer $l$ are connected by $\phi_{k'k}^{(l)}$ that are sparse. In particular, we can understand the meaning of each hidden unit by projecting it back to the original data space via $\left[\prod_{t=1}^{l-1} \mathbf{\Phi}^{(t)}\right] \phi_k^{(l)}$. We show in Fig. 4 a subnetwork, originating from units 16, 19, and 24 of the top hidden layer, taken from the generative network of size 128-64-32 inferred on Wiki. The semantic meaning of each topic and the connections between different topics are highly interpretable. We provide several additional topic hierarchies for Wiki in the Appendix.

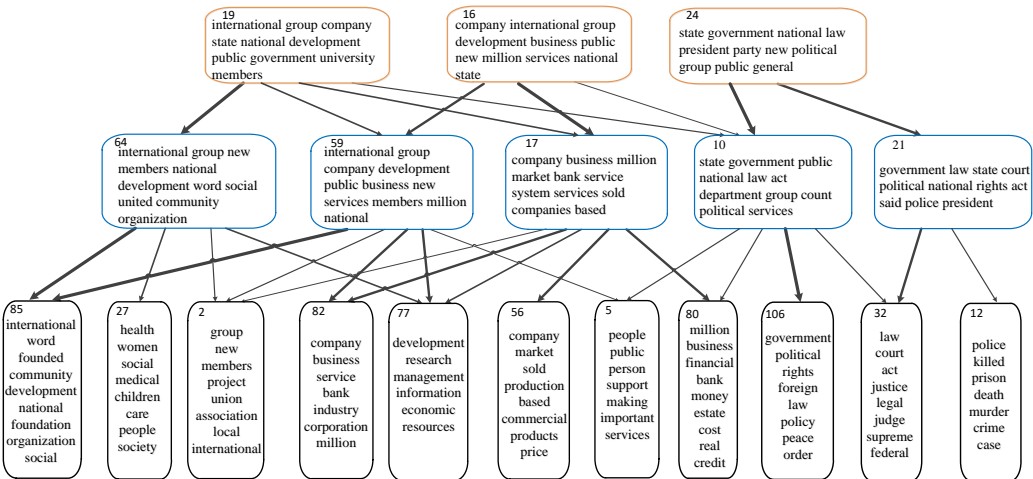

Figure 4: An example of hierarchical topics learned from Wiki by a three-hidden-layer WHAI of size 128-64-32.

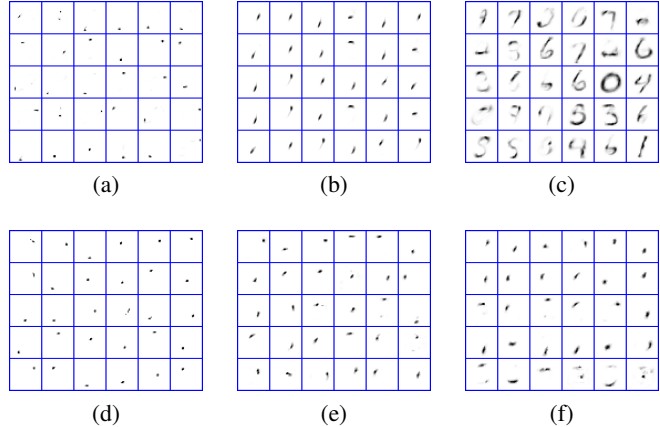

Figure 5: Learned topics on MNIST digits with a three-hidden-layer WHAI of size 128-64-32. Shown in (a)-(c) are example topics for layers 1, 2 and 3, respectively, learned with a deterministic-upward-stochastic-downward encoder, and shown in (d)-(f) are the ones learned with a deterministic-upward encoder.

To further illustrate the effectiveness of our multilayer representation in our model, we apply a three-hidden-layer WHAI to MNIST digits and present the learned dictionary atoms. We use the Poisson likelihood directly to model the MNIST digit pixel values that are nonnegative integers ranging from 0 to 255. As shown in Figs. 5a-5c, it is clear that the factors at layers one to three represent localized points, strokes, and digit components, respectively, that cover increasingly larger spatial regions. This type of hierarchical visual representation is difficult to achieve with other types of deep neural networks (Srivastava et al., 2013; Kingma & Welling, 2014; Rezende et al., 2014; Sonderby et al., 2016).

WUDVE, the inference network of WHAI, has a deterministic-upward-stochastic-downward structure, in contrast to a conventional VAE that often has a pure deterministic bottom-up structure. Here, we further visualize the importance of the stochastic-downward part of WUDVE through a simple experiment. We remove the stochastic-downward part of WUDVE shown in (6) and define the inference network as $q(\boldsymbol{\theta}_n^{(l)} \,|\, \boldsymbol{h}_n^{(l)}) = \text{Weibull}(\boldsymbol{k}_n^{(l)}, \boldsymbol{\lambda}_n^{(l)})$, in other words, we ignore the top-down information. As shown in Figs. 5d-5f, although some latent structures are learned, the hierarchical

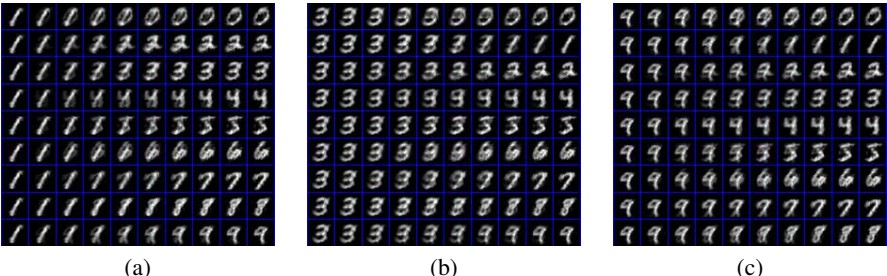

(a)  (b)  (c)

Figure 6: Latent space interpolations on the MNIST test set. Left and right columns correspond to the images generated from $z_1^{(3)}$ and $z_2^{(3)}$, and the others are generated from the latent representations interpolated linearly from $z_1^{(3)}$ to $z_2^{(3)}$.

relationships between adjacent layers almost all disappear, indicating the importance of having a stochastic-downward structure together with a deterministic-upward one in the inference network.

As a sanity check for latent representation and overfitting, we shown in Fig. 6 the latent space interpolations between the test set examples on MNIST dataset, and provide related results in the Appendix for the 20News corpus. With the 3-layer model learned before, following Dumoulin et al. (2016), we sample pairs of test set examples $x_1$ and $x_2$ and project them into $z_1^{(3)}$ and $z_2^{(3)}$. We then linearly interpolate between $z_1^{(3)}$ and $z_2^{(3)}$, and pass the intermediary points through the generative model to generate the input-space interpolations. In Fig. 6, the left and right column are the digits generated from $z_1^{(3)}$ and $z_2^{(3)}$, while the middle ones are generated from the interpolation latent space. We observe a smooth transitions between pairs of example, and intermediary images remain interpretable. In other words, the latent space the model learned is on a manifold, indicating that WHAI has learned a generalizable latent feature representation rather than concentrating its probability mass exclusively around training examples.

## 4 CONCLUSION

To infer a hierarchical latent representations of a big corpus, we develop Weibull hybrid autoencoding inference (WHAI) for deep latent Dirichlet allocation (DLDA), a deep probabilistic topic model that factorizes the observed high-dimensional count vectors under the Poisson likelihood and models the latent representation under the gamma likelihood at multiple different layers. WHAI integrates topic-layer-adaptive stochastic gradient Riemannian (TLASGR) MCMC to update the global parameters given the posterior sample of a mini-batch's local parameters, and a Weibull distribution based upward-downward variational autoencoder to infer the conditional posterior of the local parameters given the stochastically updated global parameters. The use of the Weibull distribution, which resembles the gamma distribution and has a simple reparameterization, makes one part of the evidence lower bound (ELBO) analytic, and makes it efficient to compute the gradient of the non-analytic part of the ELBO with respect to the parameters of the inference network. Moving beyond deep models and inference procedures based on Gaussian latent variables, WHAI provides posterior samples for both the global parameters of the generative model and these of the inference network, yields highly interpretable multilayer latent document representation, is scalable to a big training corpus due to the use of a stochastic-gradient MCMC, and is fast in out-of-sample prediction due to the use of an inference network. Compelling experimental results on big text corpora demonstrate the advantages of WHAI in both quantitative and qualitative analysis.

## 5 ACKNOWLEDGE

This work is partially supported by the Fund for Foreign Scholars in University Research and Teaching Programs (the 111 Project) (No. B18039), the Thousand Young Talent Program of China, NSFC (61771361) , NSFC for Distinguished Young Scholars (61525105), and Innovation Fund of International Exchange Program for Graduate Student of Xidian University.

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

## A  HIERARCHICAL TOPICS LEARNED FROM WIKI

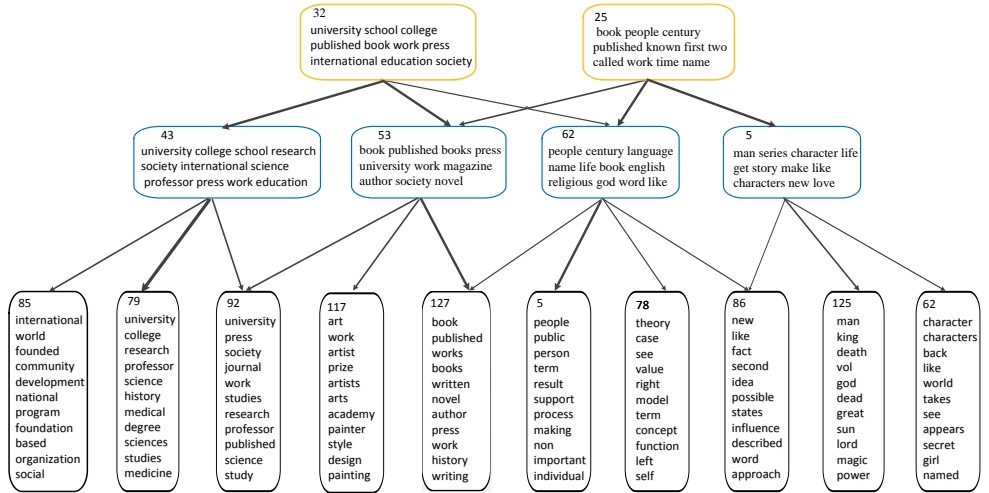

Figure 7: An example of hierarchical topics learned from Wiki by a three-hidden-layer WHAI of size 128-64-32.

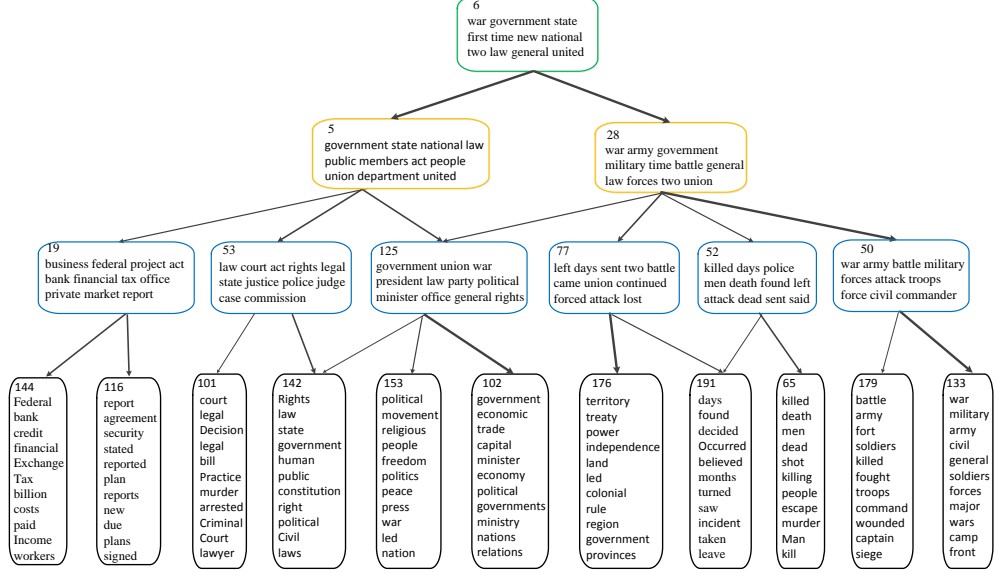

Figure 8: An example of hierarchical topics learned from Wiki by a four-hidden-layer WHAI of size 256-128-64-32.

# B  MANIFOLD ON DOCUMENTS

**From a sci.medicine document to an eci.space one**

1. com, writes, article, edu, medical, pitt, pain, blood, disease, doctor, medicine, treatment, patients, health, ibm

2. com, writes, article, edu, space, medical, pitt, pain, blood, disease, doctor, data, treatment, patients, health

3. space, com, writes, article, edu, data, medical, launch, earth, states, blood, moon, disease, satellite, medicine,

4. space, data, com, writes, article, edu, launch, earth, states, moon, satellite, shuttle, nasa, price, lunar

5. space, data, launch, earth, states, moon, satellite, case, com, shuttle, price, nasa, price, lunar, writes,

6. space, data, launch, earth, states, moon, orbit, satellite, case, shuttle, price, nasa, system, lunar, spacecraft

**From a alt.atheism document to a soc.religion.christian one**

1. god, just, want, moral, believe, religion, atheists, atheism, christian, make, atheist, good, say, bible, faith

2. god, just, want, believe, jesus, christian, atheists, bible, atheism faith, say, make, religious, christians, atheist

3. god, jesus, just, faith, believe, christian, bible, want, church, say, religion, moral, lord, world, writes

4. god, jesus, faith, just, bible, church, christ, believe, say, writes, lord, religion, world, want, sin

5. god, jesus, faith, church, christ, bible, christian, say, write, lord, believe, truth, world, human, holy

6. god, jesus, faith, church, christ, bible, writes, say, christian, lord, sin, human, father, spirit, truth

**From a com.graphics document to a comp.sys.ibm.pc.hardware one**

1. image, color, windows, files, image, thanks, jpeg, gif, card, bit, window, win, help, colors, format

2. image, windows, color, files, card, images, jpeg, thanks, gif, bit, window, win, colors, monitor, program

3. windows, image, color, card, files, gov, writes, nasa, article, images, program, jpeg, vidio, display, monitor

4. windows, gov, writes, nasa, article, card, going, program, image, color, memory, files, software, know, screen

5. gov, windows, writes, nasa, article, going, dos, card, memory, know, display, says, screen, work, ram

6. gov, writes, nasa, windows, article, going, dos, program, card, memory, software, says, ram, work, running

