# OpenReview forum: "WHAI: Weibull Hybrid Autoencoding Inference for Deep Topic Modeling"
_ICLR.cc/2018/Conference — Accept (Poster)_

### Official Review · AnonReviewer2 · 2017-11-28

**Rating:** 6
**Confidence:** 4

**Review:**

The authors develop a hybrid amortized variational inference MCMC inference
framework for deep latent Dirichlet allocation. Their model consists of a stack of
 gamma factorization layers with a Poisson layer at the bottom. They amortize
inference at the observation level using a Weibull approximation. The structure
of the inference network mimics the MCMC sampler for this model. Finally they
use MCMC to infer the parameters shared across data. A couple of questions:

1) How effective are the MCMC steps at mixing? It looks like this approach helps a
bit with local optima?

2) The gamma distribution can be reparameterized via its rejection sampler

@InProceedings{pmlr-v54-naesseth17a,
  title = 	 {{Reparameterization Gradients through Acceptance-Rejection Sampling Algorithms}},
  author = 	 {Christian Naesseth and Francisco Ruiz and Scott Linderman and David Blei},
  booktitle = 	 {Proceedings of the 20th International Conference on Artificial Intelligence and Statistics},
  pages = 	 {489--498},
  year = 	 {2017}
}

I think some of the motivation for the Weibull is weakened by this work. Maybe a
comparison is in order?

3) Analytic KL divergence can be good or bad. It depends on the correlation between
the gradients of the stochastic KL divergence and the stochastic log-likelihood

4) One of the original motivations for DLDA was that the augmentation scheme
removed the need for most non-conjugate inference. However, this approach doesn't
use that directly. Thus, it seems more similar to inference procedure in deep exponential
families. Was the structure of the inference network proposed here crucial?

5) How much like a Weibull do you expect the posterior to be? This seems unclear.

---

> ### Author Response · Authors · 2017-12-18
> **Detailed response to Reviewer 3's questions, with newly added comparison to gamma + RSVI for hybrid autoencoding inference**
>
> We thank Reviewer 3 for his/her feedback. We have made revisions accordingly, with the main changes highlighted in blue. Below please find our detailed response.
>
> Q1: How effective are the MCMC steps at mixing? It looks like this approach helps a bit with local optima?
>
> A: The MCMC steps of DLDA-WHAI are quite effective, as demonstrated in Fig. 3 by its clearly faster convergence in comparison to DLDA-WAI, which uses SGD. We also think that WHAI helps escape local optima. In DLDA-WAI, we use the same method with Srivastava & Sutton (2017) to realize the simplex constraint on \Phi^{(l)}, which  needs a well-tuned regularization parameter on \Phi^{(l)} in order to achieve a good performance and meaningful topics, especially for a deep model with two or more hidden layers. Thus, the MCMC steps of WHAI also help eliminate sensitive tuning parameters.
>
> Q2: The gamma distribution can be reparameterized via its rejection sampler called rejection sampling variational inference (RSVI) proposed in Naesseth et al. (2017). I think some of the motivation for the Weibull is weakened by this work. Maybe a comparison is in order?
>
> A: Thank you very much for the suggestion. Indeed, RSVI is an excellent method that lets us apply reparameterization to a much wider class of variational distribution, including approximately reparameterizing the gamma distribution. We have now included it into the revised paper, with the corresponding algorithm developed under RSVI referred to as gamma hybrid autoencoding inference (GHAI).
>
> Although RSVI is an attractive technique that is very general, as shown in the updated Table 2, DLDA-GHAI clearly underperforms DLDA-WHAI, suggesting that the potential benefits of using the gamma over the Weibull are overshadowed by the approximations made in RSVI, where the accepted noise for reparameterization are correlated with the gamma distribution parameters and some additional uniform random numbers are needed. Please see our discussion in Section 2.4 and added results on gamma hybrid autoencoding inference (GHAI) in Section 3.1 for more details.
>
> Q3: Analytic KL divergence can be good or bad. It depends on the correlation between the gradients of the stochastic KL divergence and the stochastic log-likelihood.
>
> A: We agree with your comment. Our results suggest that using Weibull provides both analytic KL and a good guidance of the gradient with respect to the ELBO for our deep model.
>
> Q4: One of the original motivations for DLDA was that the augmentation scheme removed the need for most non-conjugate inference. However, this approach doesn’t use that directly. Thus, it seems more similar to inference procedure in deep exponential families. Was the structure of the inference network proposed here crucial?
>
> A: The deterministic-upward and stochastic-downward structure of the inference network is crucial to obtain good and interpretable results for a deep model. For example, as shown in Fig. 5, if we remove all downward links of the inference network, the inferred latent factors become much less meaningful, and as shown in Table 1, GHAI-Independent and WHAI-Independent fail to improve as the model goes deeper.
>
> This particular structure is inspired by the upward-downward Gibbs sampler of DLDA developed with data augmentation. Although DLDA can be considered as a special case of deep exponential families (DEFs), it is distinct from the other existing DEFs in having an upward-downward Gibbs sampling. For a DEF that does not have an upward-downward Gibbs sampler, we find it difficult to come up with an appropriately structured inference network that stochastically connects different hidden layers. That is probably why existing DEFs except for DLDA almost always use mean-filed variational inference, without explicit information propagation between layers.
>
> We also note this particular structure may also be generalized to develop a deterministic-upward and stochastic-downward inference network for other models such as sigmoid belief network.
>
> Q5: How much like a Weibull do you expect the posterior to be? This seems unclear.
>
> A: We choose the Weibull distribution to approximate the gamma distributed conditional posterior shown in Equation 5 in the paper. With DLDA-Gibbs or DLDA-TLASGR, in general, the shape parameters in Equation 5 are found to be neither too close to zero nor too large, thus, as suggested by Fig. 1, we expect the Weibull to well approximate the gamma distributed conditional posteriors.

---

> > ### Public Comment · ~Christian_A_Naesseth1 · 2018-03-06
> > **regarding rsvi**
> >
> > Very interesting comparison results between Weibull and Gamma for this model! In general I would expect this to be model and data specific: in some cases the posterior is better approximated by a Gamma, and in others Weibull.
> >
> > Just a small comment regarding RSVI, with B=1 the probability of accepting is always higher than 0.95. If you set B=4 it will be higher than 0.99, making the difference between proposal and target very small. For this B you might even achieve better performance by just omitting the extra score function term, which is most likely negligible when compared to the reparameterization term.

---

### Official Review · AnonReviewer1 · 2017-11-28
**a deep Poisson model**

**Rating:** 6
**Confidence:** 2

**Review:**

The paper presents a deep Poisson model where the last layer is the vector of word counts generated by a vector Poisson. This is parameterized by a matrix vector product, and the vector in this parameterizeation is itself generated by a vector Gamma with a matrix-vector parameterization. From there the vectors are all Gammas with matrix-vector parameterizations in a typical deep setup.

While the model is reasonable, the purpose was not clear to me. If only the last layer generates a document, then what use is the deep structure? For example, learning hierarchical topics as in Figure 4 doesn't seem so useful here since only the last layer matters. Also, since no input is being mapped to an output, what does going deeper mean? It doesn't look like any linear mapping is being learned from the input to output spaces, so ultimately the document itself is coming from a simple linear Poisson model just like LDA and other non-deep methods.

The experiments are otherwise thorough and convincing that quantitative performance is improved over previous attempts at the problem.

---

> ### Author Response · Authors · 2017-12-18
> **Clarifications for why it is desired to have a deep structure for topic modeling.**
>
> We thank Reviewer 2 for his/her comments and questions. We have made revisions accordingly and highlighted our main changes in blue.
>
> If we only use a single hidden layer, then \{\theta_{nk}\}_{k}, the weights of the topics in document n, follow independent gamma distributions in the prior. By going deep, we are able to construct a much more expressive hierarchical prior distribution, whose marginal is designed to capture the correlations between different topics at multiple hidden layers. From the viewpoint of deep learning, our multilayer deep generative model provides a distributed representation of the data, with a higher layer capturing an increasingly more general concept. Empirically, our experiments consistently show that making a model deeper leads to improved performance.
> In Figure 4, without the deep structure, the inferred first-layer topics will have worse qualities, and their relationships will become difficult to understand.
>
> Our deep model is a deep generative model that has multiple stochastic layers. It is unsupervised trained to learn how to transform the gamma random noises injected at multiple different hidden layers to generate the correlated topic weights at the first layer, which are further multiplied with the learned topics as the Poisson rates to generate high-dimensional count vectors under the Poisson distribution. Thus, even though the Poisson layer is the same between a shallow model and a deep one, the latter has a much more sophisticated mechanism to generate (correlated) topic weights at the first layer, and infers a network to understand the complex relationships between different topics at multiple different levels.

---

### Official Review · AnonReviewer3 · 2017-11-29
**WHAI: WEIBULL HYBRID AUTOENCODING INFERENCE FOR DEEP TOPIC MODELING**

**Rating:** 5
**Confidence:** 4

**Review:**

The authors propose a hybrid Bayesian inference approach for deep topic models that integrates stochastic gradient MCMC for global parameters and Weibull-based multilayer variational autoencoders (VAEs) for local parameters. The decoding arm of the VAE consists of deep latent Dirichlet allocation, and an upward-downward structure for the encoder. Gamma distributions are approximated as Weibull distributions since the Kullback-Leibler divergence is known and samples can be efficiently drawn from a transformation of samples from a uniform distribution.

The results in Table 1 are concerning for several reasons, i) the proposed approach underperfroms DLDA-Gibbs and DLDA-TLASGR. ii) The authors point to the scalability of the mini-batch-based algorithms, however, although more expensive, DLDA-Gibbs, is not prohibitive given results for Wikipedia are provided. iii) The proposed approach is certainly faster at test time, however, it is not clear to me in which settings such speed (compared to Gibbs) would be needed, provided the unsupervised nature of the task at hand. iv) It is not clear to me why there is no test-time difference between WAI and WHAI, considering that in the latter, global parameters are sampled via stochastic-gradient MCMC. One possible explanation being that during test time, the approach does not use samples from W but rather a summary of them, say posterior means, in which case, it defeats the purpose of sampling from global parameters, which may explain why WAI and WHAI perform about the same in the 3 datasets considered.

- \Phi is in a subset of R_+, in fact, columns of \Phi are in the P_0-dimensional simplex.
- \Phi should have K_1 columns not K.
- The first paragraph in Page 5 is very confusing because h is introduced before explicitly connecting it to k and \lambda. Also, if k = \lambda, why introduce different notations?

---

> ### Author Response · Authors · 2017-12-18
> **A detailed response to Reviewer 1's concerns on the results in Table 1.**
>
> We thank Reviewer 1 for his/her comments and suggestions. We have revised the paper accordingly, with the revised/added texts highlighted in blue. Below please find our response to Reviewer 1’s concerns on the results in Table 1.
>
> Q1: The proposed approach underperforms DLDA-Gibbs and DLDA-TLASGR.
>
> A: Measured by per-heldout-word perpelexity, DLDA-WHAI only slightly underperforms DLDA-Gibbs and DLDA-TLASGR. However, DLDA-WHAI is substantially faster than both DLDA-Gibbs and DLDA-TLASGR for sampling a multi-layer latent representation of a testing document given a sample of the global parameters. This is because to sample a latent representation for a document from the conditional posterior, while both DLDA-Gibbs and DLDA-TLASGR often require quite a few Gibbs sampling iterations, DLDA-WHAI requires only a single deterministic-upward projection, followed by a single stochastic-downward random draw.
>
> Q2: The authors point to the scalability of the mini-batch-based algorithms, however, although more expensive, DLDA-Gibbs, is not prohibitive given results for Wikipedia are provided.
>
> A: DLDA-Gibbs, which needs to process all documents in each iteration, requires a memory that is large enough to store all data and local parameters. For the 10-million-document Wiki dataset, a PC with 32G memory actually failed to run, and we had to use a workstation with 64G memory to obtain the results for DLDA-Gibbs. Note that 64G memory will ultimately become insufficient as one further increases the data size. Even if one can have a machine that may forever increase its memory to satisfy the need of DLDA-Gibbs, Fig. 3 shows that DLDA-Gibbs needs much longer time before achieving satisfying results, while a mini-batch based algorithm has already made substantial progress before DLDA-Gibbs even finishes a single iteration.
>
> Q3: The proposed approach is certainly faster at test time, however, it is not clear to me in which settings such speed (compared to Gibbs) would be needed, provided the unsupervised nature of the task at hand.
>
> A: The purpose of our probabilistic generative model is to extract the topics and learn the multilayer latent representation under these topics in an unsupervised manner. Being able to extract the latent representation of a test document with a low computational cost makes it attractive to be used in a wide variety of real applications. For example, to process a large number of incoming documents in real time, to process documents in mobile devices with low power consumptions, to quickly identify the key topics of a news article/blog post and recommend it to relevant users, and to rapidly extract the topic-proportion vector of a document and use it to retrieve related documents. Please see Srivastava & Sutton (2017) for additional discussions on the importance of fast inference for topic models.
>
> Q4: It is not clear to me why there is no test-time difference between WAI and WHAI, considering that in the latter, global parameters are sampled via stochastic-gradient MCMC. One possible explanation being that during test time, the approach does not use samples from W but rather a summary of them, say posterior means, in which case, it defeats the purpose of sampling from global parameters, which may explain why WAI and WHAI perform about the same in the 3 datasets considered.
>
> A: As shown in Fig. 3, WHAI converges faster than WAI, although the final perplexity obtained by averaging over collected samples are similar. While they share the same inference for the neural-network parameters of the auto-encoder, WHAI uses TLASGR-MCMC while WAI uses SGD to update \Phi^{(l)}. We have added Mandt, Hoffman & Blei (2017) to support the practice of using SGD to obtain the approximate posterior samples of W. At the test time, both WHAI and WAI use the same number of samples of the global parameters, and use the auto-encoder of the same structure to generate the latent representation of a test document under each global-parameter sample, which is why WHAI and WAI have the same test time.
>
> Newly added reference: S. Mandt, M. D. Hoffman, and D. M. Blei. Stochastic gradient descent as approximate Bayesian inference. arXiv:1704.04289, to appear in Journal of Machine Learning Research, 2017.
>
> Our answer to the other comments: we have now clearly specified the simplex constraint on the columns of \Phi and clearly defined the neural networks for k, \lambda, and h.

---

### Decision · Program_Chairs · 2018-01-29
**ICLR 2018 Conference Acceptance Decision**

**Decision:**

Accept (Poster)

**Comment:**

The paper proposes a new approach for scalable training of deep topic models based on amortized inference for the local parameters and stochastic-gradient MCMC for the global ones.  The key aspect of the method involves using Weibull  distributions (instead of Gammas) to model the variational posteriors over the local parameters, enabling the use of the reparameterization trick. The resulting methods perform slightly worse that the Gibbs-sampling-based approaches but are much faster at test time. Amortized inference has already been applied to topic models, but the use of Weibull posteriors proposed here appears novel. However, there seems to be no clear advantage to using stochastic-gradient MCMC instead of vanilla SGD to infer the global parameters, so the value of this aspect of WHAI unclear.